# Low birth weight and associated factors among HIV positive and negative mothers delivered in northwest Amhara region referral hospitals, Ethiopia,2020 a comparative crossectional study

**Elsa Awoke Fentie**◯ *, **Hedija Yenus Yeshita, Moges Muluneh Bokie**◯

Department of Reproductive Health, Institute of Public Health, University of Gondar, Gondar, Ethiopia

* elsaawoke91@gmail.com

**Data Availability Statement:** All the relevant data are within the manuscript and its supporting information files.

## Abstract

### Background

Even though pregnancy does not affect HIV infection progression, HIV affects the pregnancy outcome. Maternal HIV infection has many untoward effects which include low birth weight which is the major cause of neonatal, infant, and under-five mortality. However, there is controversy and limited information about the effect of HIV status on birth weight around the world including Ethiopia. Therefore, this study aimed to compare the prevalence of LBW and their associated factors among HIV+ and HIV- mothers delivered in Northwest Amhara region referral hospitals.

### Method

A comparative crossectional study was conducted from September 2016 to September 2019. A simple random sampling technique was used to select 474 participants. Data were collected from the mothers' chart by using a data extraction sheet and then entered into Epidata and exported into SPSS for analysis. Independent variables with p-values < 0.2 in the bivariable analysis were entered into multivariable logistic regression models with backward logistic regressions method to control confounders and identify the factor.

### Result

The overall prevalence of LBW was 13.9% (95% CI:10.8%-17.1%). The prevalence was higher among HIV+ 17.7% (95% CI:14.1%-22.8%) than HIV- mothers 10.1% (95% CI:6.3%—13.8%). CD4 count < 200 cells/mm3 [AOR 3.2, 95%CI (1.05, 9.84)] and between 200–350 cells/mm3 [2.81, 95% CI (1,08, 7.28)], Mothers with MUAC <23 cm [AOR 3.39, 95% CI (1.41, 8.18)] and gestational age <37 weeks [AOR 7.34, 95% CI (3.02,17.80)] were significantly associated with LBW in HIV+ mothers. While, rural residence [AOR 3.93,95% CI (1.356,11.40)], PROM during current pregnancy [AOR 4.96, 95% CI (2.55, 15.83)] and

**Funding:** The author(s) received no specific fund for this work.

**Competing interests:** The authors have declared that no competing interest exist.

**Abbreviations:** AIDS, Acquire Immune Deficiency Syndrome; ANC, Antenatal Care; AOR, Adjusted Odd Ratio; APH, Anti Partum Hemorrhage; ART, Anti-Retroviral Treatment; BMI, Body Mass Index; CI, Confidence Interval; COR, Crude Odds Ratio; DM, Diabetes Mellitus; EDHS, Ethiopian Demographic Survey; HAART, Highly Active Anti-Retroviral Therapy; HIV, Human Immune Virus; HTN, Hypertension; IQ, Intelligent Quiescent; LBW, Low Birth Weight; LNMP, Last Normal Menstrual Period; MUAC, Mid Upper Arm Circumference; NVP, Nevirapine; PI, Protease Inhibitor; PIH, Pregnancy Induced Hypertension; PMTCT, Prevention of Mother to Child Transmission; PROM, Premature Rupture of Membrane; PTB, Pre Term Birth; RTI, Reproductive Tract Infection; SD, Standard Deviation; SPSS, Statical Package for Social Science; STI, Sexually transmitted infection; UOG, University of Gondar; UTI, Urinary Tract Infection; WHO, World Health Organization.

gestational age <37 week [AOR 8.21, 95% CI (2.60, 25.89)] were significantly associated with LBW in HIV negative mothers.

## Conclusion

The prevalence of LBW was significantly higher among HIV+ mothers as compared to HIV—mothers and this study suggests to emphasize nutritional supplementation of HIV positive mothers, needs to focus on nutritional counseling during ANC/PMTCT follow up and encourage HIV positive mothers to delay their pregnancy until their immune status improve.

## Introduction

Since the beginning of the epidemic, more than 74.9 million people have been infected with HIV globally. By the end of 2018 37.9 million people were living with HIV and 82% of pregnant women's residing with HIV had access to Anti-Retroviral Treatment (ART) to stop transmission of HIV to their baby globally [1]. According to the World health organization (WHO) report, the African region remains the most severely affected area particularly East and Southern Africa [2]. In Ethiopia by 2018, nearly 690,000 people were living with HIV/AIDS and the adult prevalence rate was 1% and the incidence rate was 0.24% per year [3].

The prevalence of HIV increased from 2005 to 2011 in most of the regions Ethiopia (1.4% in 2005 and 1.5% in 2011), including Dire Dawa, Addis Ababa, Gambella, South Nations, SNNPR, Benishangule Gumuz, and Somali. On the other hand, in the later 5 years duration, 2011 to 2016, the prevalence was decreased in all of the administrative regions, decreasing from 1.5% to 0.9% [4]. the pooled prevalence of HIV in pregnant women in Ethiopia was 5.74%. Besides, subgroup analysis was done based on different regions of Ethiopia and there is significant variation in HIV prevalence between regions, the pooled prevalence among subgroups indicated 9.50% in Amhara, 4.80% in Addis Ababa, 2.14% in SNNP, and 4.48% in the Oromia region [5]. This pooled estimate is higher than the national HIV prevalence among the general population of Ethiopia.

Pregnancy does not affect HIV infection progression [6]. However, HIV affects birth outcomes. Several studies in Africa and other countries revealed that HIV-positive mothers had a high risk of adverse birth outcomes, including, low birth weight babies, stillbirths, and preterm birth [7–10]. In contrast, a study done in India showed no differences between HIV-infected and uninfected mothers with respect to obstetric and birth outcomes [11].

The birth rate of Ethiopia in 2019 was **32.109** births per 1000 people, and it was **declined** from 2018 by **1.42%** [12]. Low birth weight (LBW) is defined as a birth weighing below 2,500 grams irrespective of gestational age. Globally 20.5 million live births suffered from LBW in 2015 and almost half of them in Southern Asia which is 9.8 million and about one-quarter of all LBW newborns are in sub-Saharan Africa [13]. According to the Ethiopian demography health survey (EDHS) 2016, the proportion of births weighing less than 2.5 kg at birth was 13% [14]. LBW is a major risk factor for neonatal, infant, and under-five mortality [15, 16]. Likewise, these children experience more morbidity, both in the short and long term, including suffered from stunted growth, lower Intelligent Quiescent (IQ) and the consequences of low birthweight continue into adulthood, increasing the risk of adult-onset chronic conditions such as obesity and diabetes [17, 18].

Studies revealed that mothers in the teenage age group, age >30 years, and residing from the rural area had a greater risk of having LBW baby [19, 20]. Different studies showed that

behavioral factors like alcohol consumption, cigarette smoking, and chewing Khat during pregnancy also increases the risks of LBW [20, 21]. Being anemic during the pregnancy, history of chronic medical conditions, having had urinary tract infection (UTI) during pregnancy, malaria infection, Untreated reproductive tract infection (RTIs), bad obstetric history, and pregnancy-related complications increase the risk of delivering LBW baby [9, 20, 22]. The magnitude of LBW among mothers with mid-upper arm circumference (MUAC) less than 23cm was higher when compared with those with MUAC greater than 23cm [22, 23]. Moreover, mothers whose body mass index (BMI) was below 18.5 were at high risk of LBW [9, 19]

The baseline maternal CD4 counts below 200 cells/mm3, maternal HIV status, maternal exposure to highly active antiretroviral therapy (HAART), advanced-stage HIV disease, intrauterine HIV transmission, and viral load ≥20 000copies/ml are factors influencing the occurrence of adverse birth outcomes [9, 19, 21, 24].

Even though there is an advancement of medical technologies, improvement in utilization of antenatal care (ANC), Prevention of Mother to Child Transmission (PMTCT), and institutional delivery LBW is still a public health problem in Ethiopia. However, there are few studies done related to birth outcomes in HIV-infected women in Ethiopia and even the existing ones can't conclude regarding the effect of HIV on birth outcomes because the studies focus only on the HIV+ pregnant women and there is no HIV- comparison group. Therefore, this study aims to assess the LBW and associated factors among HIV+ and HIV- women in northwest Amhara region referral hospitals.

## Methods and materials

### Study design, period, and area

A hospital-based comparative cross-sectional study was conducted among mothers delivered from September 2016 to September 2019 in northwest Amhara region referral hospitals and the data was extracted from March 3 to April 4 and May 5- May 18/ 2020. In the northwest part of the Amhara region, there are 3 referral hospitals such as; University of Gondar comprehensive and specialized Hospital (UoGCSH), Felege Hiwot comprehensive, and specialized hospital (FHCS), and Debre Markos referral hospital. Each referral hospital's catchment population is estimated to be 5–7 million people. the annual average number of births in each hospital is 6000 per year. The overall incidence rate of new HIV infection from 2015 to 2018 in the Amhara region was 6.9 per1000 tested population. The incidence rate was higher in females (4.1 per1000 population) than in males (2.84 per1000 population) [25]. The Ethiopian government started to implement Option B+ (initiation of antiretroviral therapy for all pregnant mothers) PMTCT service in 2013. Since then, the Option B+ treatment option has been launched in all health facilities and provided without fee. According to the operational plan, under Option B+, all HIV+ pregnant mothers will receive triple antiretroviral therapy (ART) drugs and will continue the treatment for the rest of their lives. Focused antenatal care is provided in those hospitals and this care recognize all pregnant women are at risk of complication, therefore, it provides safe, simple, and cost-effective intervention to all pregnant women to maintain normal pregnancies, save lives by preventing complications or early detection, and treatment of complications.

### Population

All mothers delivered from September 2016 to September 2019 in northwest Amhara region referral hospitals were considered to study participants. All mothers delivered from September 2016 –September 2019 in northwest Amhara region referral hospitals with a gestational age of 28 weeks and above were included in the study. However, Mothers who had unknown or

unreliable last normal menstrual period (LNMP) with the absence of ultrasound evidence, a mother with unrecorded birth weight.

## Sample size determination, Sampling procedure, and study variables

The required sample size was determined by using a double population proportion formula by taking the required statistical assumptions $Z\alpha/2 = 1.96$, power ($\beta$) = 0.84, r = ratio of $N_2$ to $N_1$ which is taken as 1, P1 = Proportion of LBW among HIV positive mothers = 21.4% [9], P2 = Proportion of LBW among HIV negative mothers = 11.9% [26]. The total final estimated sample size was 474 The calculated sample size was distributed equally among the two populations(N1 = 237, N2 = 237). A simple random sampling technique was used to select the study participants. The total sample size was proportionally allocated for the three hospitals. Medical Record Number (MRN) of study participants was filtered first from the logbook of each referral Hospital according to their delivery time and HIV status then we gave a serial number for the remaining participants and select each record for our study using a computer-generated random number. LBW was the outcome variable and sociodemographic factor (age, residence, educational status, history of substance use, including alcohol drinking and smoking), Maternal medical and obstetric related factors (anemia, chronic medical disease, UTI, PIH, APH, and PROM, previous history of abortion, previous history of stillbirth, previous history of low birth weight, parity, gravidity), Nutrition-related factor (nutritional counseling during ANC, iron and folic acid supplementation during pregnancy, pre-pregnancy BMI, MUAC,), HIV related status (CD4 count, viral load, WHO clinical stage of the disease, initiation of ART, time of initiation of ART, time of diagnosis with HIV, types of ARV) were independent variables.

## Definition of variables

**Low birth weight.** A birth weight < 2500 gram irrespective of gestational age [22].

**UTI.** Defined as a documented clinical/laboratory diagnosis of UTI any time during the pregnancy [27].

**Gestational age.** determined by clinicians during antenatal visit or delivery using the last normal menstrual period (LNMP), or early ultrasound evidence [9].

**APH.** defined as any vaginal bleeding in the mother after 28 weeks of gestation as documented in the records by the attending clinician [27].

**PIH.** defined clinically as a blood pressure of >140/90 mmHg after 20 weeks of gestation with or without proteinuria and/or edema as diagnosed and documented by the attending clinician [27].

**Stillbirth.** Dead birth after 28th week of gestation and before the expulsion from the uterus [28].

**Anemia.** Documented Hgb level below 11gm/dl laboratory diagnosis [27].

## Data collection instrument, data collection procedures, and quality assurance

Data were collected from mothers' charts using a structured checklist prepared in English. The data extraction sheet is designed based on study objectives and developed by reviewing national and international literature and by observing charts. Three supervisors having a second degree in clinical midwifery and six data collectors having a first degree in midwifery were involved in the data collection process. A 5% preliminary chart review was conducted in the Gondar university comprehensive and specialized hospital before the actual data collection and amendments were considered based on the result of a preliminary chart review. Data collectors and supervisors were trained for one day regarding the technique and data collection process by the principal

investigator before the actual data collection. Frequent and timely supervision of data collectors was undertaken by the supervisor and principal investigator. The collected data was checked out for its completeness during data collection by the principal investigator and supervisor.

## Data processing and analysis

Data were coded and then entered, edited, and cleaned using EPI data version 4.6 and exported to SPSS 25 statistical software for analysis. we manage missing data by using replacement technique if less than 20% of value are missed in one variable (E.g. we managed by replacement technique pre-pregnancy BMI, MUAC, ANC follow up, iron duration, anemia.)but if more than 20% of values missed in one variable we discard the variables (E.g. we discard marital status, educational status, occupational status, substance abuse, pregnancy status, pre-pregnancy weight). The outcome variable was dichotomized and coded as '0' and '1', representing those who have birth weight $> = 2500$ K.g. and have $< 2500$ K.g. respectively. Descriptive statistics were used to describe the socio-demographic characteristics of the respondents, the magnitude of LBW of HIV+ and HIV- women. Text and tables were used to present the findings. The binary logistic regression model was used to assess the association between dependent and independent variables. Variables with a p-value of less than 0.2 in bivariable logistic regression were considered for multivariable logistic regression analysis. In the multivariable analysis, a P-value of less than 0.05 and an odds ratio with 95% CI were used to declare the presence and the strength of association between the independent and outcome variable. Before conducting the multivariable logistic regression model multicollinearity was checked using variance inflation factor (VIF) and there is no multicollinearity between independent variables. The Hosmer and Lemeshow test was used to diagnose the model fitness and the model were adequate.

## Ethical approval and consent to participate

Ethical approval was obtained ethical review committee of the Institute of public health on behalf of the Institutional Review Board (IRB) of University of Gondar. Permission was obtained from the clinical director of each hospital. Since this study uses secondary data to ensure confidentiality Personal Identifiers Were not used on the data collection form, and All data were kept strictly confidential and used only for the study purposes.

## Results

### Sociodemographic characteristics and nutrition-related factors

During the period of review, 237 HIV+ and 237 HIV- mothers and their newborn characteristics were extracted from ANC and delivery registers and analyzed. Among a total of 474 delivered mothers participated in the study, 80.2% of HIV+ mothers were in the age group 20–34 years with a mean age 30.13 (S.D ±4.5) and 86.9% of HIV- mothers were in the age group 20–34 with a mean age of 27.05 (S.D ± 4.7). Regarding residents of the mother, 97.0% were HIV + and 69.2% of HIV—women were urban residents.

During antenatal care follow up 99.2% of HIV+ and 97% of HIV- mothers were counseled about dietary intake. More than three-quarters (84%) of HIV+ mothers had MUAC ≥23cm (Table 1).

### Medical, obstetric related characteristics, Pregnancy and labor-related complications of the mothers

In this study, twenty-five (10.5%) HIV + mothers and eight (3.4%) HIV- mothers were diagnosed with anemia during the current pregnancy. Among the study participants, 82.7% of

**Table 1. Sociodemographic and nutrition-related characteristics of mothers delivered in Northwest Amhara regional state referral hospitals (N = 474).**

| Variable | categories | HIV positive | | HIV negative | |
|---|---|---|---|---|---|
| | | Frequency | % | Frequency | % |
| **Age** | < 20 | 1 | 0.4 | 8 | 3.4 |
| | 20–34 | 190 | 80.2 | 206 | 86.9 |
| | > = 35 | 46 | 19.4 | 23 | 9.7 |
| **Resident** | Urban | 230 | 97.0 | 157 | 66.2 |
| | Rural | 7 | 3.0 | 80 | 33.8 |
| **Iron intake** | No | 2 | 0.8 | 7 | 3.0 |
| | Yes | 235 | 99.2 | 230 | 97.0 |
| **Duration of iron** | < 3 months | 145 | 61.2 | 155 | 65.4 |
| **Intake** | ≥3 months | 90 | 38.0 | 75 | 31.6 |
| **Nutritional counseling** | No | 2 | 0.8 | 7 | 3 |
| | Yes | 235 | 99.2 | 230 | 97 |
| **Pre pregnancy BMI** | <18.5 | 30 | 15.3 | | |
| | ≥18.5 | 166 | 84.7 | | |
| **MUAC** | <23cm | 38 | 16.0 | | |
| | ≥23cm | 199 | 84.0 | | |

HIV+ and 59.5% of HIV- mothers had a history of multi-gravidas. Almost all (99.2%) of HIV + and 97.5% of HIV- mothers had ANC follow up and of which 63.3% of HIV+ and 37.6%of HIV- mothers had four and above ANC follow up respectively. Common pregnancy-related complications retrieved from records were PROM (18.1% Vs 13.9% for HIV- and HIV + respectively). More than three-quarters of labor in HIV+ and HIV- mothers (85.6% and 81.9% respectively) were initiated spontaneously. About 77.6% HIV + and 63.7% of HIV—mothers current deliveries were spontaneous vaginal delivery (Table 2).

## HIV related characteristics of the mother

The majority (81.0%) of HIV + mothers know their HIV status before pregnancy and 99.6% were in WHO clinical stage one. Among HIV+ mothers, almost all (99.2%) were on ART, of which 81.0% started HAART before pregnancy and all of them had good drug adherence. more than half of (73.0%) HIV + mothers CD4 count were $\geq 351mm^3$ (Table 3).

## Prevalence of LBW

The finding of this study showed that the overall magnitude of LBW among mothers delivered in west Amhara regional state referral hospitals was 13.9% (95% CI:10.8%-17.1%) with a mean birth weight of 2938.2 gram (S. D±439.1). Based on the mother's HIV status the magnitude of LBW was higher in HIV- mothers, which is 17.7% (95% CI:14.1%-22.8%) with the mean birth weight of 2837.97 gram (SD±464.885), while among HIV- mothers the prevalence of LBW was 10.1% (95% CI:6.3%—13.8%) with the mean birth weight of 3033.3 gram (SD±395.1) (Table 4)

## Determinants of LBW

Multivariable analysis result of HIV-mothers showed that women with CD4 count less than 200 cells/mm3 and between 200–350 cells/mm3 were 3 times [AOR 3.2, 95%CI (1.05, 9.84)] and 2.8 times [AOR 2.81, 95% CI (1,08, 7.28)] more likely to have LBW baby respectively compared with those have CD4 count greater than 350 cells/mm3. Mothers with MUAC <23 cm

**Table 2.** Medical, obstetric related characteristics, Pregnancy and labor-related complications of the mothers delivered in Northwest Amhara regional state referral hospitals (N = 474).

| Variable | categories | HIV positive | | HIV negative | |
|---|---|---|---|---|---|
| | | Frequency | % | frequency | % |
| **Anemia** | No | 212 | 89.5 | 229 | 96.6 |
| | Yes | 25 | 10.5 | 8 | 3.4 |
| **History of HTN** | No | 235 | 99.2 | 237 | 100 |
| | Yes | 2 | 0.8 | 0 | 0 |
| **History of DM** | No | 237 | 100 | 236 | 99.6 |
| | Yes | 0 | 0 | 1 | 0.4 |
| **STI during current pregnancy** | No | 231 | 97.5 | 232 | 97.9 |
| | Yes | 6 | 2.5 | 5 | 2.1 |
| **Type of STI** | Syphilis | 5 | 2.1 | 4 | 1.7 |
| | HBSg | 1 | 0.4 | 1 | 0.4 |
| **UTI** | No | 222 | 93.7 | 233 | 98.3 |
| | Yes | 15 | 6.3 | 4 | 1.7 |
| **Gravidity** | Primigravida | 41 | 17.3 | 96 | 40.5 |
| | Multigravida | 196 | 82.7 | 141 | 59.5 |
| **History of LBW** | No | 230 | 97.0 | 233 | 98.3 |
| | Yes | 7 | 3.0 | 4 | 1.7 |
| **History of spontaneous abortion** | No | 202 | 85.2 | 210 | 88.6 |
| | Yes | 35 | 14.8 | 27 | 11.4 |
| **History of stillbirth** | No | 223 | 94.1 | 226 | 95.4 |
| | Yes | 14 | 5.9 | 11 | 4.6 |
| **ANC follow up** | No | 2 | 0.8 | 6 | 2.5 |
| | Yes | 235 | 99.2 | 231 | 97.5 |
| **Number of ANC** | <4 | 85 | 35.9 | 142 | 61.5 |
| | > = 4 | 150 | 63.3 | 89 | 37.6 |
| **PIH** | No | 223 | 94.1 | 197 | 83.1 |
| | Yes | 14 | 5.9 | 40 | 16.9 |
| **PROM** | No | 204 | 86.1 | 194 | 81.9 |
| | Yes | 33 | 13.9 | 43 | 18.1 |
| **APH** | No | 234 | 98.7 | 224 | 94.5 |
| | Yes | 3 | 1.3 | 13 | 5.5 |
| **Malpresentation** | No | 225 | 94.9 | 213 | 89.9 |
| | Yes | 12 | 5.1 | 24 | 10.1 |
| **Prolonged labor** | No | 209 | 88.2 | 172 | 72.6 |
| | Yes | 11 | 4.6 | 41 | 17.3 |
| | Elective C/S | 17 | 7.2 | 24 | 10.1 |
| **Labor status** | Induced | 17 | 7.2 | 19 | 8.0 |
| | Spontaneous | 203 | 85.6 | 194 | 81.9 |
| | Elective C/S | 17 | 7.2 | 24 | 10.1 |
| **Mode of delivery** | SVD | 184 | 77.6 | 151 | 63.7 |
| | Cesarean section | 52 | 22 | 75 | 31.6 |
| | Instrumental delivery | 1 | 0.4 | 11 | 4.6 |

were 3 times [AOR 3.39, 95% CI (1.41, 8.18)] more likely to have LBW babies compared with their counterparts. Newborn babies who were delivered before the gestational age of 37 weeks were 7 times [AOR 7.34, 95% CI (3.02,17.80)] higher to become low birth weight When compared to babies born at a gestational age of 37 weeks and more (Table 5).

**Table 3. HIV related characteristics of the mother delivered in west Amhara regional state referral hospitals N = 237).**

| Variable | Categories | HIV positive | |
|---|---|---|---|
| | | Frequency | % |
| Time of HIV diagnosis | before pregnancy | 192 | 81.0 |
| | during pregnancy | 43 | 18.2 |
| | during delivery | 2 | 0.8 |
| ART intervention | No | 2 | 0.8 |
| | Yes | 235 | 99.2 |
| Time HAART initiated | before pregnancy | 192 | 81 |
| | during pregnancy | 43 | 19 |
| HAART regimen | 1c | 62 | 26.3 |
| | 1d | 14 | 6.0 |
| | 1e | 143 | 60.9 |
| | Other | 16 | 6.8 |
| HAART adherence | Good | 235 | 100 |
| | Fair | 0 | 0 |
| | Poor | 0 | 0 |
| PMTCT follow up | No | 0 | 0 |
| | Yes | 237 | 100 |
| WHO clinical stage | stage 1 | 236 | 99.6 |
| | stage 2 | 1 | 0.4 |
| CD4 count | <200 | 23 | 9.7 |
| | 200–350 | 41 | 17.3 |
| | > = 351 | 173 | 73.0 |
| viral load | TND | 215 | 91.6 |
| | < 1000 | 17 | 7.2 |
| | > = 1000 | 3 | 1.2 |

1c: AZT+3TC+NVP, 1d: AZT+3TC+EFV, 1e: TDF+3TC+EFV, TND: target not detected.

Multivariable analysis result of HIV- mothers revealed that mothers living in a rural area were 4 [AOR 3.93,95% CI (1.356,11.40)] times more likely to have LBW babies when compared to those mothers who live in the urban area. the odds of delivering LBW babies among mothers who had PROM during current pregnancy were 5 [AOR 4.96, 95% CI (2.55, 15.83)] times higher than their counterparts. Newborn babies who were delivered before the gestational age of 37 weeks were 8 [AOR 8.21, 95% CI (2.60, 25.89)] times higher to become low birth weight When compared to babies born at a gestational age of 37 weeks and more (Table 6).

## Discussion

This study compares the prevalence of LBW in HIV + and HIV—mothers delivered in northwest Amhara region referral hospitals. The prevalence of LBW in this study shows difference between the two target populations. In which LBW among HIV + was 17.7% (95% CI:14.1%-22.8%) whereas it was 10.1% (95% CI:6.3%—13.8%). in HIV- mothers. This finding is supported by the study conducted in Nigeria 48.3%adverse pregnancy outcome occurs in HIV + women compared to 30.3% adverse pregnancy outcomes in the HIV-women and in which low birth weight was 9.4% Vs 3.3% in HIV+ and HIV—mothers respectively [29].

A study done in Calabar teaching hospital, Nigeria, stated that there is a higher proportion of LBW among HIV+ mothers(21.7%) compared with HIV- mothers(14.4%)(8). Moreover,

**Table 4. Birth outcome in Northwest Amhara regional state referral hospitals (N = 474).**

| Variable | Categories | HIV positive | | HIV negative | | Overall | |
|---|---|---|---|---|---|---|---|
| | | Frequency | % | Frequency | % | Frequency | % |
| Birth outcome | Alive | 235 | 99.2 | 234 | 98.7 | 469 | 98.9 |
| | Still birth | 2 | 0.8 | 3 | 1.3 | 5 | 1.1 |
| Sex of newborn | Male | 125 | 52.7 | 132 | 55.7 | 257 | 54.22 |
| | Female | 112 | 47.3 | 105 | 44.3 | 217 | 45.78 |
| Birth weight | <2500gram | 42 | 17.7 | 24 | 10.1 | 66 | 13.9 |
| | ≥2500gram | 195 | 82.3 | 213 | 89.9 | 408 | 86.1 |

the finding of this study also supported by a study done in Ghana based on maternal HIV infection status, the prevalence of LBW was higher among HIV infected mothers (22.5% Vs 14.1%), and a study done in South Africa revealed that LBW was higher in HIV+ mothers (14% Vs 9%) [30, 31]. A meta-analysis of cohort studies revealed that the prevalence of LBW among HIV infected women ranged from 3.4 to 56.0% and 2.5 to 36.9% in HIV uninfected women [24].

The discrepancy of birth weight among HIV+ and HIV- mothers might be due to compromised immune system of the mother increase the risk of opportunistic infections, which contributed to the occurrence of adverse birth outcomes [24] or it might be due to HAART particularly NVP-based HAART increased risk of preterm birth compared with EFV-based HAART [32] or it might be due to undernutrition secondary to chronic medical conditions (HIV) [9], in which malnourished mothers are highly prone to having LBW baby.

However, a study done in India showed that no differences between HIV+ and HIV— mothers with respect to obstetric and birth outcomes [11]. This might be due to the study excluding severely ill women which may lead to adverse birth outcomes or it might be due to the good nutritional status of the mother that may lead to good birth outcomes.

**Table 5. Bivariable and multivariable logistic regressions of factors associated with LBW among HIV positive mothers.**

| Variables | Response | LBW | | COR (95% CI) | AOR (95%) |
|---|---|---|---|---|---|
| | | No | Yes | | |
| **Anemia** | No | 181 | 31 | 1 | 1 |
| | Yes | 14 | 11 | 4.58(1.91,11.03) | 2.67(0.97, 7.34) |
| **PROM** | No | 173 | 31 | 1 | 1 |
| | Yes | 22 | 11 | 2.79(1.23,6.32) | 1.30(0.45, 3.78) |
| **Number of** | > = 4 | 131 | 19 | 1 | 1 |
| **ANC** | < 4 | 63 | 22 | 2.41(1.25,4.77) | 1.26(0.49, 8.35) |
| **CD4 count** | > = 351 | 151 | 25 | 1 | 1 |
| | 200–350 | 30 | 7 | 2.52(1.11,5.73) | **2.81(1.08, 7.28)** * |
| | <200 | 14 | 10 | 4.341(1.73, 10.77) | **3.22(1.05,9.84)** * |
| **MUAC** | > = 23cm | 172 | 27 | 1 | 1 |
| | <23 cm | 23 | 15 | 4.15(1.71, 11.40) | **3.39(1.41, 8.18)** * |
| **Duration of** | > = 3 months | 79 | 11 | 1 | 1 |
| **iron intake** | < 3months | 115 | 30 | 1.874(0.89, 3.96) | 1.68(0.71, 3.99) |
| **Gestational** | > = 37 weeks | 179 | 24 | 1 | 1 |
| **Age** | < 37 weeks | 16 | 18 | 8.34(3.78,18.62) | **7.34(3.02,17.80)** * |

*P-value <0.05.

**Table 6. Bivariable and multivariable logistic regressions of factors associated with LBW among HIV negative mothers.**

| Variables | Response | LBW | | COR (95%CI) | AOR (95% CI) |
|---|---|---|---|---|---|
| | | No | Yes | | |
| **Residence** | Urban | 148 | 9 | 1 | 1 |
| | Rural | 65 | 15 | 3.79(1.58, 9,12) | **3.93(1.356,11.40)** * |
| **PROM** | No | 181 | 13 | 1 | 1 |
| | Yes | 32 | 11 | 4.79(2.97, 11.62) | **4.96(2.55, 15**.83) * |
| **PIH** | No | 183 | 14 | 1 | 1 |
| | Yes | 30 | 10 | 4.36(1.77, 9.70) | 3.34(0.97, 10.5) |
| **Number of ANC** | > = 4 | 85 | 4 | 1 | 1 |
| | < 4 | 123 | 19 | 3.28(1.078, 9.99) | 1.68(0.30, 9.35) |
| **Duration of** | > = 3months | 72 | 3 | 1 | 1 |
| **Iron intake** | < 3 months | 135 | 20 | 3.56(1.02, 12.37) | 3.578(0.77,15.57) |
| **Gestational** | > = 37 | 190 | 10 | 1 | 1 |
| **Age** | < 37 | 23 | 14 | 11.57(4.61, 29.01) | **8.21(2.60, 25.89)** * |

The odds of being LBW in babies born before a gestational age of 37 weeks was 7 times higher in HIV+ and 8 times higher in HIV- mothers when compared to babies born at a gestational age of 37 weeks and more. This finding is consistent with studies done in Gondar [9, 33, 34] Tigray [35], China [21], Iran[36], and Bangladeshi [37]. This could be because the baby deliver before completion of their normal physical development in the womb which leads to LBW [35]

In this study mothers who had a CD4 count below 200 cells/mm3 and between 200–350 cells/mm3 were 3.2- and 2.8-times higher risk of having LBW respectively as compared to those with CD4 level above 350 cells/mm3. This result is similar to the studies carried out in Northwest Ethiopia public hospitals [9], China, Tanzania [38] and Nigeria [29] this might be due to compromised immune system of the mother may increase the risk of opportunistic infections which affect mothers health, nutritional status and intrauterine fetal growth [9].

HIV+ mothers who had MUAC <23 cm had 3.4 folds higher risk of LBW compared to those who had MUAC ≥ 23cm. This finding is in line with studies done in Dessie referral hospital [22] and Kersa district, southern Ethiopia [23]. This might be due to the intergenerational effect of malnutrition [21], which means undernutrition of the mother may increase the risk of intrauterine growth restriction [35].

HIV- mothers residing in rural areas were 3.9 times more likely to have LBW babies compared with mothers residing from urban. This finding is in line with studies done in Tigray [35], Mekelle hospital [39], and Hosana [20]. This could be because mothers who live in rural areas have lower access to medical services and have poor awareness about health, and nutrition [35]. Rural residence of the mother had an effect on birth weight, which increases the risk of neonatal, infant, and under-five mortality, therefore, the concerned body should work on increasing medical service access and improve their awareness about nutrition.

The likelihood of having LBW baby among HIV- Mothers who had PROM during their current pregnancy was found to be 4.9 times higher compared to those who did not have PROM. This result is supported by studies conducted in South Africa [40] and Kenya [41]. This could be due to provider-initiated early termination of pregnancy because of pregnancy-related complications which lead to LBW [42] or it might be due to that labor will spontaneously initiate within a week after preterm PROM [43] which leads to delivery of a baby before completion of normal physical development.

HIV status of the mother was significantly associated with the LBW among all mothers, the odds of delivering LBW baby among HIV positive mothers were 4(AOR 4.2 95% CI [1.89–

9.43]) times higher than negative mothers. this finding is in line with studies done in Gondar university hospital [27, 44], Nigeria [29], USA [45], South Africa [46], and meta-analysis conducted in developed and developing countries [24]. This might be due to compromised immune system of the mother increase the risk of opportunistic infection, which contributes to the occurrence of LBW [24, 47] or another possible explanation is ART drugs especially PI-based increase prematurity [48] and NVP-based HAART resulted in an increased risk of preterm birth [33], which might in turn cause LBW. The result of this study indicated that being HIV positive increases the risk of delivering LBW, therefore, it is better to emphasize HIV prevention.

## Limitation of study

Since this study is hospital-based, it doesn't include mothers who gave birth at home. In addition, data used were secondary there may be bias and incomplete information's.

## Conclusion

In our study, the prevalence of LBW was significantly higher among HIV+ mothers than HIV —mothers. CD4 count $<$ 200cells/mm$^3$ and between 200–350 cells/mm$^3$, MUAC $<$23cm, and gestational age $<$ 37 weeks were important contributing factors for LBW among HIV + mothers. On the other hand, Rural residence, PROM during the current pregnancy, and gestational age $<$ 37 weeks were factors contributing to LBW among HIV- mothers. Therefore, the nutrition program needs to emphasize activities that improve the nutritional status of HIV+ mothers. Health care providers need to focus on nutritional counseling during ANC/PMTCT follow-up and encourage HIV + mothers to delay their pregnancy until their immune status improves.

## Supporting information

**S1 File.**
(ZIP)

## Acknowledgments

The authors acknowledge the University of Gondar for securing ethical clearance for this study. We would like to extend our gratitude to Specialized and referral Hospitals found in the northwest Amhara regional state, and data collectors.

## Author Contributions

**Conceptualization:** Elsa Awoke Fentie.

**Data curation:** Elsa Awoke Fentie.

**Formal analysis:** Elsa Awoke Fentie, Hedija Yenus Yeshita, Moges Muluneh Bokie.

**Investigation:** Elsa Awoke Fentie.

**Methodology:** Elsa Awoke Fentie.

**Resources:** Elsa Awoke Fentie.

**Supervision:** Hedija Yenus Yeshita, Moges Muluneh Bokie.

**Validation:** Elsa Awoke Fentie.

**Writing – original draft:** Elsa Awoke Fentie.

**Writing – review & editing:** Elsa Awoke Fentie, Hedija Yenus Yeshita, Moges Muluneh Bokie.

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
