## [Decision Letter · Decision Letter 0]

10 Jun 2021

PONE-D-20-40543

Prevalence of Low birth weight and associated factors among HIV positive and negative mothers delivered in northwest Amhara region referral hospitals, Ethiopia,2020 A comparative crossectional study

PLOS ONE

Dear Dr. Fentie,

Thank you for submitting your manuscript to PLOS ONE. After careful consideration, we feel that it has merit but does not fully meet PLOS ONE’s publication criteria as it currently stands. Therefore, we invite you to submit a revised version of the manuscript that addresses the points raised during the review process.

The reviewers have identified several aspects of the methods and analysis in your study that require substantial clarification. Please ensure that you thoroughly address all of the points raised by the reviewers when preparing your revised manuscript.

We look forward to receiving your revised manuscript.

Kind regards,

Jamie Males

Staff Editor

PLOS ONE

Journal Requirements:

the author(s) received no specific fund for this work

 [xxx]

Reviewers' comments:

Reviewer's Responses to Questions

**Comments to the Author**

1. Is the manuscript technically sound, and do the data support the conclusions?

Reviewer #1: Yes

Reviewer #2: Partly

2. Has the statistical analysis been performed appropriately and rigorously? 

Reviewer #1: I Don't Know

Reviewer #2: No

3. Have the authors made all data underlying the findings in their manuscript fully available?

Reviewer #1: Yes

Reviewer #2: Yes

4. Is the manuscript presented in an intelligible fashion and written in standard English?

Reviewer #1: No

Reviewer #2: No

5. Review Comments to the Author

Reviewer #1: This is an interesting paper which shows important differences in the birth weights of infants born to HIV positive mothers compared to HIV negative mothers.

The paper needs to be edited and adjusted by somebody with experience in writing journal articles. There are numerous grammatical and language issues which need to be corrected. These will not be specifically highlighted here as there too many to mention.

Introduction:

i) It should be made clear that the 74.9 million is worldwide

ii) More information about number of births in Ethiopia per year. Home birth rate etc.

iii) More information about HIV in Ethiopia would be useful. Is it increasing/decreasing? What % of people are on ARVs etc. Incidence should be quoted as 0.24% per year.

iv) The paragraph about LBW in the world needs to be reworked to make it flow better

v) what is chewing 'chat'? Most readers would not know this.

Methods and Materials

i) It's confusing what is meant by data was extracted from March 3 to April 4 and May 5 - May 18.

Results

This could be shortened. Some results are irrelevant (eg: Rh factor) and some tables could be combined. Results that are easily interpreted in the table do not necessarily need to be mentioned in the text also, although it is understandable that some should be emphasized.

Prevalence of LBW.

Perhaps a box and whisker plot to break the monotony of the tables. Don't need so many decimal places (eg: 439.065)

Why include Still births? What is the benefit?

It is also confusing why preterm infants were included. They are obviously much more likely to be LBW but not necessarily growth restricted. Growth restriction is the whole focus of the paper. So these should either be excluded or determined if they are growth restricted using growth charts for gestational age.

No baby HIV results? It would be important to know the transmission rate. HIV positive infants should be a different baby from an HIV exposed but uninfected infant.

Discussion

Some of the points mentioned above need to be mentioned in the discussion. What about emerging evidence that ARVs may cause hypertension in the mothers and therefore may produce smaller babies?

There are some studies from South Africa comparing unexposed to exposed but negative infants. These could also be included.

So all in all, an interesting study, but needs a lot of editing and revision.

Reviewer #2: • There is a lot of effort in collecting medical chart data from an African hospital. So, the authors are commended for pursuing this endeavor. Also, having a comparison group is a plus.

• In the introduction section, the authors make the case that there is controversy in whether contributes to LBW. While some studies have shown no association between HIV and LBW, there have been one systematic review and another meta-analysis on this topic, so the purpose of the investigation and the resulting findings are not new. See for example (Xiao, Peng-Lei, et al. 2015; 15. Brocklehurst, Peter, and Rebecca French 1998). It is important the authors recognize this and summarize the findings of those studies in the manuscript. They are cited but not discussed. Same can be said about the discussion section where the study is compared to other studies.

• The purpose of the study should be expanded. Currently, it sounds like their overall goal is to investigate HIV’s association with LBW. However, a huge part of the study is looking at risk factors for LBW among mothers living with HIV.

• To be able to somewhat confidently conclude that HIV is associated with LBW in their population, statistically, they should use propensity score matching to account for selection bias into the two groups. Currently, the prevalence of LBW is provided for each group, with an associated 95% CI. It is unclear what is driving these differences, since there is such an imbalance in the confounding variables. In addition, it is also not clear where that 95% CI statistic is coming from. Just computing the prevalence is not adequate since the sample is not a randomized selection (that is those who are HIV+ and those who are not—and not necessarily the sample of participants from the hospital)

• The prevalence of HIV is also driven by context—(For example rich countries vs. low income countries, differences in access to ART, and underlying nutritional status). The sample is largely urban and drawn from three tertiary hospitals. Tertiary hospitals see more at risk patients than non-tertiary hospitals, including more mothers living with HIV. As a result, the prevalence of LBW among the two populations could be driven by that. Relatedly, since the study was not nationally representative and only done in three tertiary hospitals, the title of the study should reflect that, so that it is not misleading as a nationally representative study. Moreover, there has to be more detail about the setting. What is the HIV prevalence among pregnant women? What proportion has access to treatment? What PMTCT services are offered?

• There are so many tables in the results section and they can be combined. For example, the demographic variables can be combined.

• Given that they are looking at so many variables, the analysis should account for multi-collinearity, using variance inflation factor.

• Missing data and missing folders are a big problem with medical chart reviews. Without disclosing how much folders were missing and how much data was also missing, is hard to have confidence in the numbers and results. The number of folders would affect the denominator in calculating LBW. This information should be provided for mothers living with and those without HIV.

• The HIV field has moved away from labelling mothers as HIV+ and HIV- to people centered labels like mothers living with HIV.

Introduction

• Antiretroviral therapy access play a significant role in prevalence of LBW and transmission of HIV. The background should discuss this relationship between access and LBW

• In justifying the study, the authors indicate that there have been previous studies conducted in Ethiopia, but they have lacked a comparison group. None of these studies are cited.

Methods

• Are the PMTCT free at the three hospitals?

• What PMTCT model does the hospital provide? Option B? Option B+?

• What is the prevalence of HIV among pregnant women? How does it compare to the general population?

• What is the justification for the 28 weeks criteria for inclusion and exclusion.

• What is the HIV prevalence at these hospitals? It should be included.

• What is the total number of births there? What is the standard of care for the patients. Readers need some context, since they may not all be familiar with the country.

• A key problem with relying on medical records is missing data and folders. The authors should provide details about missing folders and data and describe how that was addressed in the analysis.

• The authors on page 12 lists variables that were treated as independent variables. Wouldn’t those be confounding/covariates variables, since they are comparing HIV to non-HIV+ mothers?

• For the variables, the authors should consider describing the types of measurement variables—that is how they were treated in the analysis (categorical, continuous, etc)).

• Several of the variables in the Tables are not described in the methods, example stillbirth

• There are also several clinical acronyms in the tables and methods that are not defined

Results

• The number of tables are quite excessive. Table 1-3 could be combined.

• The results are impacted but the concerns raised earlier

6. PLOS authors have the option to publish the peer review history of their article (what does this mean?). If published, this will include your full peer review and any attached files.

Reviewer #1: **Yes: **Lloyd Tooke

Reviewer #2: No

---

## [Author Response · Author response to Decision Letter 0]

12 Aug 2021

Authors’ response to concerns of Editor and Reviewers

Dear Editor and Reviewers, we the authors of this article would be very happy to convey our deepest gratitude for your immense contribution - rigorous editorial and review concerns for which we will go one by one to make our manuscript suitable for publication in PLOS ONE journal. Therefore, we are going to respond the Editor, Reviewer #1 and Reviewer #2 concerns respectively as presented hereunder:

Editor concerns and Authors’ responses

Editor concern: “1. Please ensure that your manuscript meets PLOS ONE's style requirements, including those for file naming.”

Authors’ response: Dear Editor, in all parts of the revised manuscript, we have used PLOS ONE’s requirements for publication of manuscripts. All the changes have been presented in a manuscript with track changes and without track changes. 

 Editor concern: “2 We note that you have provided funding information that is not currently declared in your Funding Statement. However, funding information should not appear in the Acknowledgments section or other areas of your manuscript. We will only publish funding information present in the Funding Statement section of the online submission form. 

Please remove any funding-related text from the manuscript and let us know how you would like to update your Funding Statement. Currently, please include your amended statements within your cover letter; we will change the online submission form on your behalf.

Authors’ response: Dear Editor, in the revised manuscript, the acknowledgment section of the manuscript is restated, in fact we extend our gratitude to university of Gondar for the contribution of securing ethical clearance fee free because of this we wrongly understand the contribution as financial support. However, there is no direct financial support from university of Gondar. Therefore, all the changes have been presented in a manuscript with track changes and without track changes.

Editor concern: “3 We note that you have indicated that data from this study are available upon request. PLOS only allows data to be available upon request if there are legal or ethical restrictions on sharing data publicly. 

Authors’ response: Dear Editor, in the revised manuscript, we have been updated data availability and materials, all the relevant data are within the manuscript and its supporting information files. Therefore, all the changes have been included in a manuscript with track changes and without track changes, as well as in the cover letter. And data attached as supporting files.

Editor concern: “4 PLOS requires an ORCID iD for the corresponding author in Editorial Manager on papers submitted after December 6th, 2016. Please ensure that you have an ORCID iD and that it is validated in Editorial Manager. To do this, go to ‘Update my Information’ (in the upper left-hand corner of the main menu), and click on the Fetch/Validate link next to the ORCID field. This will take you to the ORCID site and allow you to create a new iD or authenticate a pre-existing iD in Editorial Manager. 

Authors’ response: Dear Editor, have an ORCID iD and that it is validated by Editorial Manager

Editor concern: “5 Your ethics statement should only appear in the Methods section of your manuscript. If your ethics statement is written in any section besides the Methods, please move it to the Methods section and delete it from any other section. Please ensure that your ethics statement is included in your manuscript, as the ethics statement entered into the online submission form will not be published alongside your manuscript.

Authors’ response: Dear Editor, in the revised manuscript, the ethical statement it is being moved to the methods part only, and we have presented it in a manuscript with track changes and without track changes.

2. Reviewer #1 concerns and Authors’ responses

Reviewer concern: “1.1 The paper needs to be edited and adjusted by somebody with experience in writing journal articles. There are numerous grammatical and language issues which need to be corrected. These will not be specifically highlighted here as there too many to mention.

Authors’ response: Dear Reviewer, in the revised manuscript, we included an updated English language usage, spelling and grammar. We have given the manuscript for language edition and all the changes have been included in the manuscript with track changes and without track changes

 Reviewer concern: “1.2 in the introduction part

I. It should be made clear that the 74.9 million is worldwide

Authors’ response: Dear Reviewer, “Since the beginning of the epidemic, more than 74.9 million people have been infected HIV” in the original manuscript is corrected as “Since the beginning of the epidemic, more than 74.9 million people have been infected HIV globally” in the revised version of the manuscript with track changes and without track changes.

II. More information about number of births in Ethiopia per year. Home birth rate etc.

Authors’ response: Dear Reviewer, we include additional information regarding number of births in Ethiopia per year and home birth rate in the revised version of the manuscript with track changes and without track changes.

III. More information about HIV in Ethiopia would be useful. Is it increasing/decreasing? What % of people are on ARVs etc. Incidence should be quoted as 0.24% per year

Authors’ response: Dear Reviewer, we include additional information regarding HIV in Ethiopia Is increasing or decreasing and “the incidence rate was 0.24%” in the original manuscript is corrected as “the incidence rate was 0.24% per year”. Therefore, all the changes have been included in a manuscript with track changes and without track changes.

IV. The paragraph about LBW in the world needs to be reworked to make it flow better

Authors’ response: Dear Reviewer, the paragraph about LBW in the world in the original manuscript is rewrite in the revised version of the manuscript with track changes and without track changes.

V. what is chewing 'chat'? Most readers would not know this

Authors’ response: Dear Reviewer, the word ‘chat’ in the original manuscript is corrected as “Khat (stimulant drug)” in the revised version of the manuscript with track changes and without track changes.

Reviewer concern: “1.3 in the method part

i) It's confusing what is meant by data was extracted from March 3 to April 4 and May 5 - May 18.

Authors’ response: Dear Reviewer, due to covid 19 the data collection was interrupted that is why we write the data extraction period from March 3 to April 4 and May 5 - May 18.

Reviewer concern: “1.4 in the result part

I. This could be shortened. Some results are irrelevant (eg: Rh factor) and some tables could be combined. Results that are easily interpreted in the table do not necessarily need to be mentioned in the text also, although it is understandable that some should be emphasized

Authors’ response: Dear Reviewer, we delete irrelevant variables (e.g., Rh factor) and we combine Table 2 (Medical and obstetric related characteristics of the mothers delivered in Northwest Amhara regional state referral hospitals) and Table 3 (Pregnancy and labor-related complications of the mothers delivered in west Amhara regional state referral hospitals) in one table in the revised version of the manuscript with track changes and without track changes.

II. Why include Still births? What is the benefit?

It is also confusing why preterm infants were included. They are obviously much more likely to be LBW but not necessarily growth restricted. Growth restriction is the whole focus of the paper. So, these should either be excluded or determined if they are growth restricted using growth charts for gestational age.

Authors’ response: Dear Reviewer, we include still birth to show birth outcome is alive or dead and take it as an independent variable. we can’t exclude preterm infants because the definition of LBW is birth of baby weighting below 2,500 grams irrespective of gestational age 

III. No baby HIV results? It would be important to know the transmission rate. HIV positive infants should be a different baby from an HIV exposed but uninfected infant

Authors’ response: Dear Reviewer, we extracted this data from mothers’ chart and in the mother’s chart the babies HIV status is not recorded.

3. Reviewer #2 concerns and Authors’ responses

Reviewer concern: “2.1 in the introduction part

I. Antiretroviral therapy access plays a significant role in prevalence of LBW and transmission of HIV. The background should discuss this relationship between access and LBW

Authors’ response: Dear Reviewer,

II. In justifying the study, the authors indicate that there have been previous studies conducted in Ethiopia, but they have lacked a comparison group. None of these studies are cited

Authors’ response: Dear Reviewer, we cited the study done in Ethiopia “Kebede B, Andargie G, Gebeyehu A. Birth outcome and correlates of low birth weight and preterm delivery among infants born to HIV-infected women in public hospitals of Northwest Ethiopia. 2013” in the original manuscript 

Reviewer concern: “2.2 in the method part

I. Are the PMTCT free at the three hospitals?

Authors’ response: Dear Reviewer, the PMTCT services are provided freely in those hospitals and we include in the revised version of the manuscript with track changes and without track changes.

II. What PMTCT model does the hospital provide? Option B? Option B+?

Authors’ response: Dear Reviewer, “The Ethiopian government started to implement the Option B+ (initiation of antiretroviral therapy for all pregnant mothers) PMTCT service in 2013. Since then, the Option B+ treatment option has been launched in all PMTCT health facilities and provided without fee. According to the operational plan, under Option B+, all HIV+ pregnant mothers will receive triple antiretroviral therapy (ART) drugs and will continue the treatment for the rest of their lives” and we include in the revised version of the manuscript with track changes and without track changes.

III. What is the prevalence of HIV among pregnant women? How does it compare to the general population?

Authors’ response: Dear Reviewer, “pooled prevalence of HIV in pregnant women in Ethiopia was 5.74% and the pooled prevalence among subgroups indicated 9.50% in Amhara, 4.80% in Addis Ababa, 2.14% in SNNP and 4.48% in Oromia region. This pooled estimate is higher than the national HIV prevalence among the general population of Ethiopia”. we include this information in the revised version of the manuscript with track changes and without track changes

IV. What is the justification for the 28 weeks criteria for inclusion and exclusion?

Authors’ response: Dear Reviewer, based on Ethiopian ministry of health standard if the gestational age is less than 28 weeks it is abortion and as you know our research is done among mothers who gave birth that is why we exclude gestational age less than 28 weeks of gestation.

V. What is the HIV prevalence at these hospitals? It should be included.

Authors’ response: Dear Reviewer, we can’t get evidence HIV prevalence in each hospital but there is study on Epidemiology of HIV Infection in the Amhara Region of Ethiopia, 2015 to 2018 Surveillance Data Analysis and those hospitals found in Amhara region. The result of this study showed that the overall incidence rate of new HIV infection from 2015 to 2018 in the Amhara region was 6.9 per1000 tested population. The incidence rate was higher in females (4.1 per1000 population) than in males (2.84 per1000 population). we include this information in the revised version of the manuscript with track changes and without track changes.

VI. What is the total number of births there? What is the standard of care for the patients? Readers need some context, since they may not all be familiar with the country

Authors’ response: Dear Reviewer, the annual average number of births in each hospital is 6000 per year. Focused antenatal care is provided in those hospitals and this care recognize all pregnant women are at risk of complication, therefore, it provides safe, simple, and cost-effective intervention to all pregnant women to maintain normal pregnancies, save the lives by preventing complications or early detection and treatment of complications. Moreover, the Option B+ treatment option is provided without fee in those hospitals and we include in the revised version of the manuscript with track changes and without track changes.

VII. A key problem with relying on medical records is missing data and folders. The authors should provide details about missing folders and data and describe how that was addressed in the analysis.

Authors’ response: Dear Reviewer, we manage missing data by using replacement technique if less than 20% of value are missed in one variable ( E.g. we managed by replacement technique pre pregnancy BMI, MUAC, ANC follow up, iron duration, anemia etc.) but if more than 20% of values missed in one variable we discard the variables ( E.g. we discard marital status, educational status, occupational status, substance abuse, pregnancy status, pre pregnancy weight) and we include in the revised version of the manuscript with track changes and without track changes

VIII. The authors on page 12 lists variables that were treated as independent variables. Wouldn’t those be confounding/covariates variables, since they are comparing HIV to non-HIV+ mothers?

Authors’ response: Dear Reviewer, we try to control cofounding variables by using appropriate statical analysis, which means first we check chi-square assumption then bivariable analysis and finally we did multivariable logistic regression by using back ward logistic regression technique 

IX. For the variables, the authors should consider describing the types of measurement variables—that is how they were treated in the analysis (categorical, continuous, etc).

Authors’ response: Dear Reviewer, data processing and analysis part shows the outcome variable is categorical and treat in the analysis by using binary logistic regression model.

X. Several of the variables in the Tables are not described in the methods, example stillbirth

Authors’ response: Dear Reviewer, we try to describe variables found in the table on the method part example Stillbirth in the revised version of the manuscript with track changes and without track changes.

XI. There are also several clinical acronyms in the tables and methods that are not defined

Authors’ response: Dear Reviewer, we defined clinical acronyms that is found in the table and method (E.g., DM, HTN, STI) in the revised version of the manuscript with track changes and without track changes.

XII. Given that they are looking at so many variables, the analysis should account for multi-collinearity, using variance inflation factor.

Authors’ response: Dear Reviewer, before conducting the multivariable logistic regression model multicollinearity was checked using variable inflation factor (VIF) and there is no multicollinearity between independent variables and we include in the revised version of the manuscript with track changes and without track changes.

Reviewer concern: “2.3 in the result part

• The number of tables are quite excessive. Table 1-3 could be combined.

Authors’ response: Dear Reviewer, we combine Table 2 (Medical and obstetric related characteristics of the mothers delivered in Northwest Amhara regional state referral hospitals) and Table 3 (Pregnancy and labor-related complications of the mothers delivered in west Amhara regional state referral hospitals) in one table in the revised version of the manuscript with track changes and without track changes.

Reviewer concern: “2.4

The HIV field has moved away from labelling mothers as HIV+ and HIV- to people centered labels like mothers living with HIV.

Authors’ response: Dear Reviewer, labels like mothers living with HIV is corrected as “HIV+ and HIV- mothers” in the revised version of the manuscript with track changes and without track changes.

Reviewer concern: “2.5

• In the introduction section, the authors make the case that there is controversy in whether contributes to LBW. While some studies have shown no association between HIV and LBW, there have been one systematic review and another meta-analysis on this topic, so the purpose of the investigation and the resulting findings are not new. See for example (Xiao, Peng-Lei, et al. 2015; 15. Brocklehurst, Peter, and Rebecca French 1998). It is important the authors recognize this and summarize the findings of those studies in the manuscript. They are cited but not discussed. Same can be said about the discussion section where the study is compared to other studies.

Authors’ response: Dear Reviewer, we try to discuss the cited references on the discussion session and we include the change in the revised manuscript with track change and without track changes. Regarding to the issue raised on “the purpose of the investigation and the resulting findings are not new” yes you are right there are systematic review and metanalysis on this topic but there are few studies done related to birth outcomes in HIV-infected women in Ethiopia even the existing ones can’t show the disparity between HIV+ and HIV- mothers.

Reviewer concern: “2.6

To be able to somewhat confidently conclude that HIV is associated with LBW in their population, statistically, they should use propensity score matching to account for selection bias into the two groups. Currently, the prevalence of LBW is provided for each group, with an associated 95% CI. It is unclear what is driving these differences, since there is such an imbalance in the confounding variables. In addition, it is also not clear where that 95% CI statistic is coming from. Just computing the prevalence is not adequate since the sample is not a randomized selection (that is those who are HIV+ and those who are not—and not necessarily the sample of participants from the hospital)

Authors’ response: Dear Reviewer, to say HIV is statically associated with birth weight we do regression analysis in the overall population (which means taking both HIV positive and negative as one population) by taking HIV status as independent variable and we get being HIV+ increase the risk of delivering low birth weight baby compared with HIV negative mothers at P-value 0.000 (AOR 4.2 95% CI [1.89-9.43]). we include this change in the revised manuscript with and without track change

---

## [Decision Letter · Decision Letter 1]

26 Nov 2021

PONE-D-20-40543R1Prevalence of Low birth weight and associated factors among HIV positive and negative mothers delivered in northwest Amhara region referral hospitals, Ethiopia,2020 A comparative crossectional studyPLOS ONE

Dear Dr. Fentie,

Thank you for submitting your manuscript to PLOS ONE. After careful consideration, we feel that it has merit but does not fully meet PLOS ONE’s publication criteria as it currently stands. Therefore, we invite you to submit a revised version of the manuscript that addresses the points raised during the review process.

We look forward to receiving your revised manuscript.

Kind regards,

Grzegorz Woźniakowski, Full professor, PhD, ScD

Academic Editor

PLOS ONE

Journal Requirements:

Reviewers' comments:

Reviewer's Responses to Questions

**Comments to the Author**

1. If the authors have adequately addressed your comments raised in a previous round of review and you feel that this manuscript is now acceptable for publication, you may indicate that here to bypass the “Comments to the Author” section, enter your conflict of interest statement in the “Confidential to Editor” section, and submit your "Accept" recommendation.

Reviewer #1: (No Response)

2. Is the manuscript technically sound, and do the data support the conclusions?

Reviewer #1: Yes

3. Has the statistical analysis been performed appropriately and rigorously? 

Reviewer #1: I Don't Know

4. Have the authors made all data underlying the findings in their manuscript fully available?

Reviewer #1: Yes

5. Is the manuscript presented in an intelligible fashion and written in standard English?

Reviewer #1: Yes

6. Review Comments to the Author

Reviewer #1: This is much improved. I only have a few comments: There are still a few grammatical errors such as "is suffered from" LBW and 'Result' instead of results.

It was stated that chewing "chat" would be explained in the text as "khat (stimulant)" but this was not done.

Well done

7. PLOS authors have the option to publish the peer review history of their article (what does this mean?). If published, this will include your full peer review and any attached files.

Reviewer #1: **Yes: **Lloyd Tooke

---

## [Author Response · Author response to Decision Letter 1]

10 Jan 2022

Editor concerns and Authors’ responses

Editor concern: “1. Please review your reference list to ensure that it is complete and correct. If you have cited papers that have been retracted, please include the rationale for doing so in the manuscript text, or remove these references and replace them with relevant current references. Any changes to the reference list should be mentioned in the rebuttal letter that accompanies your revised manuscript. If you need to cite a retracted article, indicate the article’s retracted status in the References list and also include a citation and full reference for the retraction notice.

Authors’ response: Dear Editor, we reviewed our Manuscript reference list; it is complete and correct and there is no retracted cited paper 

Reviewer #1 concerns and Authors’ responses

Reviewer concern 1: There are still a few grammatical errors such as "is suffered from" LBW and 'Result' instead of results.

Authors’ response: Dear Reviewer, ‘is suffered from’ LBW and 'Result' instead of results is corrected as “suffered from” LBW and “Results” in the revised version of the manuscript with track changes and without track changes.

Reviewer concern 2: It was stated that chewing "chat" would be explained in the text as "khat (stimulant)" but this was not done

Authors’ response: Dear Reviewer, the word ‘chat’ in the original manuscript is corrected as “Khat (stimulant drug)” in the revised version of the manuscript with track changes and without track changes.

---

## [Editor Report · Decision Letter 2]

28 Jan 2022

Prevalence of Low birth weight and associated factors among HIV positive and negative mothers delivered in northwest Amhara region referral hospitals, Ethiopia,2020 A comparative crossectional study

PONE-D-20-40543R2

Dear Dr. Fentie,

We’re pleased to inform you that your manuscript has been judged scientifically suitable for publication and will be formally accepted for publication once it meets all outstanding technical requirements.

Kind regards,

Grzegorz Woźniakowski, Full professor, PhD, ScD

Academic Editor

PLOS ONE
---

## [Editor Report · Acceptance letter]

2 Feb 2022

PONE-D-20-40543R2 

Low birth weight and associated factors among HIV positive and negative mothers delivered in northwest Amhara region referral hospitals, Ethiopia,2020 A comparative crossectional study 

Dear Dr. Fentie:

I'm pleased to inform you that your manuscript has been deemed suitable for publication in PLOS ONE. Congratulations! Your manuscript is now with our production department. 

Kind regards, 

on behalf of

Prof. Grzegorz Woźniakowski 

Academic Editor

PLOS ONE